# Ceramide Transfer Protein (CERT): An Overlooked Molecular Player in Cancer

**DOI:** 10.3390/ijms222413184

**Published:** 2021-12-07

**Authors:** Long Hoa Chung, Da Liu, Xin Tracy Liu, Yanfei Qi

**Affiliations:** Centenary Institute of Cancer Medicine and Cell Biology, University of Sydney, Camperdown, NSW 2050, Australia; darren.liu@centenary.org.au (D.L.); x.liu@centenary.org.au (X.T.L.)

**Keywords:** CERT, ceramide, sphingomyelin, sphingolipid, lipid transfer protein, lipidomics, cancer

## Abstract

Sphingolipids are a class of essential lipids implicated in constructing cellular membranes and regulating nearly all cellular functions. Sphingolipid metabolic network is centered with the ceramide–sphingomyelin axis. Ceramide is well-recognized as a pro-apoptotic signal; while sphingomyelin, as the most abundant type of sphingolipids, is required for cell growth. Therefore, the balance between these two sphingolipids can be critical for cancer cell survival and functioning. Ceramide transfer protein (CERT) dictates the ratio of ceramide to sphingomyelin within the cell. It is the only lipid transfer protein that specifically delivers ceramide from the endoplasmic reticulum to the Golgi apparatus, where ceramide serves as the substrate for sphingomyelin synthesis. In the past two decades, an increasing body of evidence has suggested a critical role of CERT in cancer, but much more intensive efforts are required to draw a definite conclusion. Herein, we review all research findings of CERT, focusing on its molecular structure, cellular functions and implications in cancer. This comprehensive review of CERT will help to better understand the molecular mechanism of cancer and inspire to identify novel druggable targets.

## 1. Introduction

Cancer cells exhibit altered lipid profile and metabolism, enabling them to meet the bioenergetic, biochemical and biophysical requirements for cancer initiation, progression and metastasis [1,2]. Within cancer cell, lipids can serve as a form of stored energy, membrane constituents and signaling molecules [3]. Meanwhile, the circulating lipids represent dietary nutrient intake and impacts, regulate hormonal and immunological responses and convey interorgan communications in cancer [4,5]. With the development of high throughput lipidomics and mass-spectrometry-based lipid imaging, an exponentially increased number of cancer-related lipid changes have been identified in cancer cells, animal models and clinical cohorts [6,7,8,9]. These often uncover previously overlooked lipid species in cancer, which offer new avenues for cancer diagnosis and treatment [6,8].

Lipid metabolites are categorized into eight well-defined groups under the guidance of the International Lipid Classification and Nomenclature Committee. It includes fatty acyls, glycerolipids, glycerophospholipids, sphingolipids, sterol lipids, prenol lipids, saccharolipids and polyketides [10]. Among them, sphingolipids are a class of critical molecular players in cancer: sphingomyelin, ceramide and glycosphingolipids serve as structural lipids in the cellular membrane, where they form sphingolipid-enriched microdomains and regulate cancer cell signaling [11,12]; while ceramide, sphingosine and sphingosine 1-phosphate (S1P) can directly function as signaling molecules implicated in cancer cell proliferation, metabolism and death, inflammation, tumor angiogenesis, metastasis and drug resistance [13,14,15,16,17,18]. Notably, ceramide has been recognized as a tumor-suppressive factor in most cancer types [19,20]. In contrast, sphingomyelin is considered essential for cancer cell proliferation, migration and immune evasion [21,22]. Therefore, the ratio of ceramide to sphingomyelin might determine the outcomes of cancer development. Supporting this notion, sphingomyelinases that hydrolyze sphingomyelin to ceramide suppress cancers and improve anti-cancer therapies [23,24,25,26,27], whereas sphingomyelin synthases that convert ceramide to sphingomyelin promote cancers [28,29,30,31,32].

In addition to sphingomyelinases and sphingomyelin synthases, ceramide transfer protein (CERT), also known as goodpasture antigen-binding protein (GPBP), collagen type IV alpha-3-binding protein (COL4A3BP) or StAR-related lipid transfer protein 11 (STARD11), is another key factor in regulating ceramide/sphingomyelin ratio. CERT is the only lipid transfer protein known to transport ceramide from the endoplasmic reticulum (ER) to the *trans*-Golgi apparatus, a prerequisite step for converting ceramide into sphingomyelin [33,34]. Since the identification of its ceramide transport function, the role of CERT in cancer has been continually studied; however, the latest review specifically on this research topic dates back to 10 years ago [35]. With the development of lipidomic analyses and the advancement of structural biology, have we obtained any new knowledge on CERT in the past decade? This article intends to provide a comprehensive review on CERT with focuses on its molecular structure, cellular functions and implications in cancer.

## 2. Sphingolipid Metabolism Centered with Ceramide/Sphingomyelin Axis

### 2.1. Chemical Structure of Sphingolipids

Sphingolipids are one of the major lipid classes in eukaryotes. These lipids build on a common backbone of an aliphatic amino alcohol, referred to as the sphingoid base [36,37]. The sphingoid base was first identified in 1884 by a German chemist J.L.W Thudichum who named this structure for its enigma as the Egyptian Sphinx [38]. The sphingoid base contains one to three hydroxyl groups, designated as “m”, “d” or “t” for mono-, di- and tri-hydroxy groups, respectively [39]. In addition, the sphingoid base can be saturated, mono-unsaturated or di-unsaturated (with double bonds in either *cis*- or *trans*-configuration) [39]. The most common sphingoid base is sphingosine which was first elucidated by H.E Carter et al. in 1947 [40]. On the sphingoid base, the addition of fatty acyl chain in different lengths (C14-C28) via N-link bond and the attachment of choline, phosphate or sugar molecules via O-link bond leads to the biodiversity of sphingolipids [36,37]. The chemical structure of major sphingolipids, including sphingosine, dihydrosphingosine (dh-sphingosine), S1P, ceramide, sphingomyelin and glycosylceramide, are listed in Figure 1. A shorthand nomenclature is currently used to denote sphingolipids (detailed in [39]). For example, sphingosine contains 18-carbon (C18) with two hydroxyl groups and one *trans*-double bond at its C4 position, and thus it is denoted as 4E-d18:1; while ceramide that possesses a fatty acyl chain in 16-carbon length via the N-link to the sphingoid base is abbreviated as C18:1/C16:0 or C16:0 in short [39]. So far, 1748 curated and 3168 computationally generated sphingolipid species have been characterized [41,42]. Among these sphingolipids, ceramide is sitting at the central point of the *de novo* biosynthetic and catabolic pathways in the sphingolipid metabolic network (Figure 2). It functions as a signaling molecule in nearly all cellular events, while sphingomyelin, generated from ceramide, represents the most abundant sphingolipid subclass in cellular membranes.

### 2.2. Biosynthesis of Sphingolipids

The first committed step of the *de novo* biosynthesis of sphingolipids is generating a sphingoid base, as shown in Figure 2. In this step, serine palmitoyl-CoA transferase (SPT) mediates the condensation of two non-sphingolipid molecules, serine and fatty acyl-CoA, to form 3-keto-dh-sphingosine at the cytoplasmic face of the ER [43,44,45]. SPT is conservatively composed of two large subunits SPT1/Lcb1p and SPT2/Lcb2p in yeast and mammals [46,47]. The third subunit, SPT3 was found later, which can replace SPT2 in a complex with SPT1 [48]. More recently, SPT is found trimeric after the identification of an additional small subunit that is required for maximal enzyme activity [49]. At the cytoplasmic side of the ER, 3-keto-dh-sphingosine is further reduced by 3-keto-dh-sphingosine reductase (KDSR) into dh-sphingosine [43,50]. Subsequently, ceramide is produced via ceramide synthase (CerS, also known as longevity assurance or Lass)-mediated N-acylation of fatty acyl chain onto the sphingoid base [51,52], followed by dh-ceramide desaturase-mediated desaturation of dihydro-ceramide in the ER [53,54]. There are six mammalian CerS isoforms, CerS1-CerS6. They exhibit different fatty acyl chain preferences and distinct tissue distribution: CerS1 is mainly expressed in brain, skeletal muscle, and testis and specifically generates C18 ceramides; CerS2 is highly expressed in liver and kidney and selectively produces C22-C24 ceramides; CerS3 is seen in testis and skin and accepts C22-C26 fatty acyls; CerS4, CerS5 and CerS6 are widely expressed in tissues, but CerS4 prefers C18-C20 fatty acyl chains while CerS5 and CerS6 predominantly generate C14-C16 ceramides [13,55,56]. Ceramide is central to the sphingolipid metabolic network (Figure 2). It can be degraded into non-lipid products via the catabolic pathway and reversibly converted into complex sphingolipids, such as sphingomyelin and glycoceramides [3]. The fatty acyl chain specificity of different CerS determines the generation of not only various ceramides but also diversified complex sphingolipids.

### 2.3. Catabolism of Sphingolipids

In the sphingolipid catabolic pathway, ceramide is deacylated into sphingosine by ceramidases [3]. According to their pH optimum, five mammalian ceramidase isoforms can be categorized into acid, neutral and alkaline ceramidase. This is also related to their subcellular localization and biological functions at the subcellular level. Acid ceramidase is maturated via proteolysis, and it works to deacylate ceramide in the acidic environment of lysosomes [57,58]. Neutral ceramidase in vertebrates is localized to the plasma membrane via the O-glycosylation of its serine/threonine/proline-rich mucin box [59]. Therefore, it is vital for the regulation of sphingosine and S1P signaling across the plasma membrane [60]. Three alkaline ceramidases function in different subcellular compartments [13]. Among them, alkaline ceramidase 1 (ACER1) is an ER-residential enzyme hydrolyzing C24:0 and C24:1 ceramides [61,62], ACER2 is a Golgi-localized enzyme preferably deacylating monounsaturated C18, C20 and C24 ceramides [63,64], while ACER3 is present in both the ER and the Golgi apparatus metabolizing unsaturated long-chain ceramides and dh-ceramides [65]. In ceramidase-mediated hydrolysis, the substrate ceramides are primarily sourced from the degradation of sphingomyelin or glycoceramides via post-Golgi vesicular trafficking [66]. Following this step, sphingosine, the product of ceramidases, is phosphorylated into S1P by sphingosine kinase 1 and 2, which is subsequently lysed by S1P lyase into ethanolamine phosphate and hexadecenal in an irreversible manner [3,67,68,69]. Sphingosine kinase 1 mainly mediates S1P production at the plasma membrane [70,71], sphingosine kinase 2 dominates the same enzymatic function in the nucleus, mitochondria and ER [72,73,74,75], while S1P lyase is an ER-localized enzyme in rodents and humans [76,77,78]. In addition, sphingosine kinase 1 contributes more to the S1P production in the circulation and cancerous tissues [18,79,80]; sphingosine kinase 2 is the predominant isoform in the liver, kidney, heart, pancreatic islets and brain, where it is a critical regulator for cell metabolism, functions and survival [81,82]; whereas S1P lyase is widely expressed in all tissues [83]. Simultaneously knockout of both sphingosine kinases, but not either individual isoform, results in embryonic lethality, indicating the essential roles of S1P and functional redundancy between sphingosine kinase 1 and 2 [84]. Knockout of S1P lyase, the gatekeeper at the exit of the sphingolipid metabolic network, elevates the levels of all sphingolipids and causes death in mice by the age of 8 weeks [83,85].

### 2.4. Complex Sphingolipids

Following *de novo* biosynthesis in the ER, a large portion of ceramide is converted to complex sphingolipids. This process is featured by a reversible addition of a substitute head group is to the 1-hydroxyl position of ceramide [39]. Therefore, complex sphingolipids are categorized by their diversified head groups into sphingomyelin, ceramide phosphoethanolamine, glycoceramides, ceramide 1-phosphate, acylceramide, etc. [86,87,88,89,90].

The most abundant sphingolipid is sphingomyelin [39,60]. In sphingomyelin synthesis, the phosphorylcholine head is transferred from phosphatidylcholine to ceramide, mediated by sphingomyelin synthases [91,92,93]. Meanwhile, phosphatidylcholine turns to be diacylglycerol [87,94,95]. The conversion between ceramide and sphingomyelin is crucial to lipid cell biology. It determines the cellular content of two major membrane structural lipids, sphingomyelin and phosphatidylcholine, as well as two starring signaling lipids, ceramide and diacylglycerol. There are two sphingomyelin synthases, sphingomyelin synthase 1 and 2. The former one is localized to the *trans*-Golgi cisterna, while the latter one is present in both *trans*-Golgi and the plasma membrane [87,96,97]. The conversion of newly biosynthesized ceramide at the *trans*-Golgi contributes the majority of sphingomyelin within the cell, while sphingomyelin production at the plasma membrane represents the recycling of ceramide. Reversibly, sphingomyelin can be hydrolyzed into ceramide, mediated by sphingomyelinases (also known as sphingomyelin phosphodiesterases or SMPDs) at different subcellular organelles [98,99]. Similar to ceramidases, sphingomyelinases are classified into acid, neutral and alkaline types, according to their pH optimum [98,99]. Acid sphingomyelinase (SMPD1) is maturated via C-terminal proteolytic processing in lysosomes, where it hydrolyzes endolysosomal sphingomyelin [100]. Further N-terminal proteolysis in endolysosomes allows acid sphingomyelinase to be secreted extracellular milieu [101,102]. The secreted form of the enzyme can metabolize sphingomyelin at the plasma membrane and in plasma under a neutral pH environment [103,104]. There are four neutral sphingomyelinases, including neutral sphingomyelinase 1–3 and mitochondria-associated neutral sphingomyelinase. Among them, neutral sphingomyelinase 1 (SMPD2) is localized to the ER [105], the Golgi apparatus [105] and the nucleus [106]; neutral sphingomyelinase 2 (SMPD3) travels between the plasma membrane and the Golgi apparatus [107,108]; neutral sphingomyelinase 3 (SMPD4) resides in the ER as well as the Golgi apparatus [109]; while the subcellular localization of mitochondria-associated neutral sphingomyelinase is told from its name [110]. Alkaline sphingomyelinase is also known as nucleotide pyrophosphatase/phosphodiesterase 7, which is mainly expressed in the intestine [111]. Within the cell, this enzyme resides in endosome-like structures near the plasma membrane [111,112].

In addition to being converted into sphingomyelin, ceramide can be catalyzed by a sphingomyelin synthase 1 homologous protein, named sphingomyelin synthase related protein, to ceramide phosphoethanolamine at the luminal side of the ER [88]. However, the content of ceramide phosphoethanolamine is 300-fold less than sphingomyelin synthase 1-produced sphingomyelin [88]. Another ceramide-derived phosphosphingolipid is ceramide 1-phosphate that possesses the simplest head group as phosphate. Ceramide 1-phosphate is produced by ceramide kinase mainly at the cytosolic face of the *trans*-Golgi, which is required for eicosanoid production [113,114,115]. Ceramide 1-phosphate can be dephosphorylated by ceramide phosphatase at the plasma membrane [116]. Following sphingomyelin, the second abundant complex sphingolipid is glycosphingolipid, such as galactosylceramide and glucosylceramide. Galactosylceramide is produced with uridine diphosphate-galactose by ceramide galactosyltransferase at the luminal side of the ER [117]; while glucosylceramide is synthesized with uridine diphosphate-glucose by glucosylceramide synthase at the *cis*-Golgi [118]. Once transported from the ER to the Golgi apparatus, galactosylceramide serves as a precursor for generating more complex glycosphingolipids, such as sulfatides and ganglioside GM4 [119,120]. In contrast, glucosylceramide synthesis is the first rate-limiting step in the production of more than 3000 glycosphingolipids, including lactosylceramides, gangliosides GM1, GM2 and GM3 [39,121]. In the revered process, galactosylceramide and glucosylceramide can be hydrolyzed into ceramide by galactosylceramidase and glucocerebrosidase in lysosomes, respectively [122,123]. Recently, ceramide is found to be acylated into a neutral lipid by diacylglycerol O-acyltransferase 2 at lipid droplets [90,124].

## 3. CERT-Mediated Intracellular Sphingolipid Trafficking

### 3.1. Two Modes of Intracellular Sphingolipid Trafficking

As introduced in the previous section, enzymes that mediate sphingolipid metabolism have distinct subcellular localization and thus access different subcellular sphingolipid pools. Interestingly, ceramide is biosynthesized in the ER, but enzymes that convert ceramide to sphingomyelin or glycoceramides are localized to the Golgi apparatus. Similarly, sphingomyelin is generated by sphingomyelin synthases in the Golgi apparatus, but hydrolysis of sphingomyelin into ceramide by sphingomyelinases mainly takes place in lysosomes or the plasma membrane. This led researchers to interrogate how sphingolipids travel from their biosynthesis sites to other subcellular venues for functioning. In general, this is not completely understood. There are two modes of intracellular sphingolipid trafficking—vesicular transport and lipid transfer protein (LTP)-mediated transport (or non-vesicular transport); both contribute to the diversified distribution of sphingolipids among subcellular compartments. Sphingomyelin is delivered from the Golgi to the plasma membrane via vesicular transport, followed by further distribution to lysosomes via endocytosis [125]. Ceramide can also travel from the ER to the *cis*-Golgi in lipoprotein or protein transport vesicles [126]. In comparison, LTP-mediated intracellular sphingolipid transport is more efficient and specific. So far, a few sphingolipid-associated LTPs have been identified, including CERT (ceramide transport from the ER to the *trans*-Golgi [34], Ceramide-1-phosphate transfer protein (ceramide 1-phosphate transport from the *trans*-Golgi network to the plasma membrane and other intracellular membranes) [127], four phosphate adapter protein 2 (glucosylceramide transport from the *cis*-Golgi to the *trans*-Golgi [128] and glycolipid transfer protein (glucosylceramide redistribution) [129]. Among these, CERT is the first and best-characterized LTP. It specifically transports ceramide from the ER to the *trans*-Golgi allowing the subsequent sphingomyelin synthesis, with no impacts on ceramide transfer to the *cis*-Golgi for glycosphingolipid production other sphingolipid trafficking [34,130].

### 3.2. Schematic Structure of CERT

CERT is encoded by the *CERT1* gene, with two protein variants in 624 and 598 amino acids, referred as to CERT_L_ and CERT, respectively (Figure 3). Its alternate names include GPBP (normally refers to CERT_L_), Col4a3bp and STARD11 (both refer to the short-form). The amino acid sequence in the short-form CERT is identical to the long-form, except for the lack of the second serine-rich motif (SRM) in the middle region (MR) [131]. CERT_L_ was initially identified by Raya et al. in 1999, which is a non-conventional serine/threonine kinase often found in the extracellular matrix or associated with the plasma membrane [132,133]; in contrast, the short-form of CERT functions as a ceramide transporter shuttling between the ER and the *trans*-Golgi, which was first addressed by Hanada et al. in 2003 [34]. The CERT protein comprises an N-terminal Pleckstrin Homology (PH) domain composed of approximately 120 amino acids, a C-terminal steroidogenic acute regulatory protein (StAR)-related lipid transfer (START) domain with about 230 amino acids and an MR between them [134]. In the MR, short-form CERT possesses an SRM at amino acid 127–152 and a two phenylalanines in an acidic tract (FFAT) motif at amino acid 321–327 [134].

### 3.3. START Domain in CERT

In humans, there are 15 proteins with the START domain (named STARD1-15) [135]. CERT is the STARD11, and its START domain specifically recognizes ceramide [136,137]. In 2008, Kudo et al. resolved the crystal structure of the START domain at the C-terminus of CERT in two conformations: an apo-form at 2.2-Å resolution and ceramide-bound form at 1.4-Å, which revealed how this domain interacted with ceramide [136]. A large amphiphilic cavity located at the center of the START domain is responsible for the binding of one ceramide molecule [136]. The cavity is formed by β-strands covering three α-helices and the Ω-loops [66,136,137]. Specifically, the amphiphilicity is contributed by 26 hydrophobic and 10 polar/charged amino acid residues within the cavity [136]. The amphiphilic nature of the cavity in the START domain permits ceramide extraction, trap and discharge [136]. This physical compatibility between the lipid and the START domain is consistent with how ceramide can flip flop in between the subcellular membranes [137]. The physical and chemical properties of the START domain result in the selective receipt and transport of ceramide molecules with particular length of saturated fatty acyl chain ranging from C14 to C20 and, less efficiently, with C22 and C24 mono-unsaturated fatty acyl chain [138]. However, it has insufficient space for C24:0 [138]. Thus, the transfer of C24:0 ceramide is rarely observed [94,136,137], which might determine the composition of sphingomyelin species [66,139]. Compared with other proteins in the START family, CERT stands out as it possesses a PH domain and an FFAT. Although the START domain is solely responsible for the ceramide binding activity in CERT, it is not sufficient for the ER-to-Golgi ceramide transport in vivo [140]. CERT mutants lacking the PH domain and FFAT motif cannot transport ceramide efficiently, as evidenced by a reduced level of sphingomyelin production [34,66,138,141].

### 3.4. PH Domain in CERT

PH domain specifically binds to phosphatidylinositol phosphates (PIPs). For example, the PH domain in phospholipase 1δ binds to PI(4,5)P_2_ [142], PI(3,4,5)P_3_ recruits protein kinase B/Akt to the plasma membrane [143,144,145], OSBP exchanges lipids between the ER and the Golgi apparatus by binding to PI(4)P [144] and class III PI 3-kinase Vps34 docks via PI(3)P for its endosomal targeting [145]. The PH domain of CERT preferably binds to PI(4)P that resides in the *trans*-Golgi and the *trans*-Golgi network (TGN) [34,146]. Surface plasmon resonance analysis has confirmed that the PH domain of CERT exhibits a higher affinity to PI(4)P-bound liposomes than PI(4,5)P2-embedded liposomes [140]. However, the PI(4)P affinity of PH-CERT is much weaker than PH-OSBP [147]. In line with this, overexpression of OSBP can inhibit ceramide transfer to the Golgi by competing in binding to the PI(4)P [148]. This work indicates that the PH domain is essential for the CERT function. Indeed, G67E mutation in the PH domain abrogates PI(4)P binding as well as ceramide transport function of CERT [34,66]. Sugiki et al. revealed the three-dimensional structure of the PH domain in CERT using nuclear magnetic resonance [140]. Structurally, a basic groove was identified near the PI(4)P recognition site, which helps to orientate the PH domain for a more efficient search for PI(4)P in the Golgi [140,149]. Interestingly, the crystal structure of the PH-START complex shows that START can bind to the PH at the PI(4) binding site [150]. Disruption of the binding between START and PH reinforces the Golgi targeting of CERT [150]. These demonstrate an intramolecular regulation mechanism, fine-tuning the subcellular localization of CERT.

### 3.5. FFAT Motif in CERT

FFAT is a short peptide motif with the canonical consensus sequence of EFFDAxE, which is present in the MR between the PH and the START domains in CERT [151]. This motif is essential for the ER targeting of CERT via the physical interaction with vesicle-associated membrane protein-associated proteins (VAPs) [66,141,152]. There are two VAPs in mammals, VAP-A and VAP-B, that provide anchors for LTPs at the cytoplasmic face of the ER [152]. Other than CERT, many LTPs, such as oxysterol binding protein (OSBP) and its related proteins (ORP1, ORP2, ORP4), Nir2 and STARD3, also transport their lipid cargos in an FFAT motif-dependent manner [153]. D324A mutation in the FFAT motif of CERT results in the loss of ceramide binding and transport [141].

### 3.6. CERT-Mediated Ceramide Transport

CERT transports ceramide from the ER to the *trans*-Golgi at their membrane contact site [66,134]. It can dramatically accelerate C16 ceramide transfer from days to 10 min in a cell-free system [34]. CERT-mediated ceramide transfer determines the cellular content of both ceramide and sphingomyelin. Knockdown of CERT causes ceramide accumulation in zebrafish embryos [154] and human colon cancer cells [155], whereas overexpression of CERT reduces ceramide level in C2C12 myoblasts [156]. In contrast, CERT is critical for sphingomyelin synthesis, as it recovers sphingomyelin levels in cells deficient in this sphingolipid [34]. CERT-deficient cells show a dramatic decrease in sphingomyelin levels but still maintain the level of glucosylceramide [34,156,157]. Ablation of CERT profoundly decreases sphingomyelin level in *Drosophila melanogaster* [158] and MCF-7 breast cancer cells [159]. The embryonic cells in CERT knockout mice exhibit a reduced level of sphingomyelin in the plasma membrane and an increased level of ceramide in the ER and mitochondria [160,161]. Consistently, adeno-associated virus-mediated overexpression of CERT_L_ decreases ceramide but increases sphingomyelin in the cortex of mice [162].

### 3.7. Regulation of CERT Function

CERT protein can be regulated by phosphorylation at least in two regions. Sequential phosphorylation occurs in the SRM [163,164,165] and serine 315 within the FFAT motif [166]. Using MALDI-TOF mass spectrometry, a characteristic “S/T-X-X” repeat was identified in the SRM of CERT protein [163]. S132, the most N-terminal serine, is known as the leading phosphorylation site required for the phosphorylation of the following serine and threonine residues [163]. S132 phosphorylation by protein kinase D decreases the PI(4)P binding and ceramide transfer [164,167]. The S132A mutation leads to the hypophosphorylation of CERT, which exhibits more powerful ceramide transfer activity as compared with the wild-type protein [163]. In line with this, the S132L mutation also results in the hypophosphorylation of CERT, which is associated with an enhanced sphingomyelin production [168]. The serial phosphorylation in the SRM by casein kinase 1 γ 2 also inactivates CERT and results in a reduction in sphingomyelin level [165]. The hyperphosphorylation in SRM also hinders PH domain access to PI(4)P, and thus it dissociates CERT from the Golgi, leading to the inactivation of CERT [169]. In contrast, myriocin (specific serine palmitoyltransferase inhibitor) and methyl-β-cyclodextrin (cholesterol-depleting reagent) can dephosphorylate and activate CERT [163]. The SRM can also be dephosphorylated by protein phosphatase 2Cε [170]. Knockdown of protein phosphatase 2Cε attenuates the association between CERT and VAPA and reduces sphingomyelin levels [170]. Recently, G243R mutation outside of SRM, FFAT, START or PH, regulates the conformational change of CERT, resulting in easy access of phosphatases to the SRM [168]. Opposing to phosphor-inhibition in the SRM, phosphorylation at S315 in the FFAT motif facilitates the VAP binding (ER targeting) of the protein, leading to an enhanced ceramide transport to the Golgi [166]. Consistently, myriocin promotes the S315 phosphorylation and activate CERT [171]. However, the kinase that is responsible for S315 phosphorylation has not been identified yet.

There is a negative feedback loop in CERT-dependent sphingomyelin production. During the synthesis of sphingomyelin in the *trans*-Golgi, diacylglycerol is co-produced [172]. Diacylglycerol recruits and activates protein kinase D, which inhibits CERT function via S132 phosphorylation [164]. In addition, PI(4)P availability is crucial for the CERT function. Genetic knockdown or pharmacological inhibition of PI 4-kinase III reduces PI(4)P level at the *trans*-Golgi, and thus it inhibits ceramide transfer [148]. OSBP depletes PI(4)P at the *trans*-Golgi in exchange for cholesterol [144]. Overexpression of OSBP inhibits CERT-mediated ceramide transfer [148], whereas blocking OSBP-mediated PI(4)P/cholesterol exchange by 25-hydroxylcholesterol promotes CERT function [173]. In OSBP-mediated regulation of CERT function and sphingomyelin synthesis, PI 4-kinase IIα, but not PI 4-kinase III, is required for PI(4)P homeostasis [174]. Interestingly, sphingomyelin production triggers dephosphorylation of PI(4)P, inhibiting CERT function and limiting over-production of sphingomyelin [175]. In a lipid transfer assay, no ceramide can be extracted by CERT from vesicles made of 100% palmitoyl-sphingomyelin [176]. Furthermore, CERT can be cleaved by caspases at its D213 residue during apoptosis [177].

## 4. Biological Function of CERT and CERT_L_

CERT exerts its biological functions by the regulation of sphingomyelin and ceramide levels at different subcellular compartments. Sphingomyelin is a chemically inert lipid and acts as a major component in the outer leaflet of the plasma membrane [178]. In particular, it interacts with cholesterol to form discrete lipid-rich microdomains in the plasma membrane, providing platforms for signaling proteins anchoring and functioning [179,180]. Sphingomyelin at the plasma membrane can also facilitate endocytosis [181]. At the *trans*-Golgi, sphingomyelin is believed to determine the protein sorting [164]. As evidenced in a few studies on manipulating sphingomyelin synthases that alter sphingomyelin content, sphingomyelin is suggested to promote cell survival [182,183], proliferation [97,184,185,186] and migration [29,187]. In contrast, ceramide is well established as a pro-apoptotic signal [188,189,190] and an inducer of cell cycle arrest [190]. Thus, the CERT might be a determinant of cancer development by dictating the ratio of ceramide to sphingomyelin. In addition to regulating ceramide and sphingomyelin, CERT_L_ is also implicated in several cellular and physiological functions via physical interaction with other proteins [162,191,192]. Here we summarize the biological functions of both short and long-form of CERT (Figure 4).

### 4.1. Biological Functions of Short-Form CERT

#### 4.1.1. Embryonic Survival and Life Span

CERT is essential for embryonic survival. Knockout of CERT results in embryonic death around E11.5 from a failed cardiovascular system in mice [160]. It reduces the levels of total cellular sphingomyelin and plasma membrane-bound sphingomyelin by more than 60%, while it increases the ceramide level in the ER and mitochondria with no change in total cellular ceramide content [160]. In accord, CERT knockout embryonic cells exhibit abnormal mitochondria morphology lacking cristae, ER stress, cell cycle arrests with a delayed entry into G2/M phase and inactivation of cell survival signal protein kinase B/Akt, attributed to increased ceramide level in the ER and mitochondria [160]. Knockdown of CERT also causes embryonic lethality in zebrafish because of increased ceramide levels and apoptosis in the brain and somites [154]. Knockout of CERT in *Drosophila melanogaster* leads to early death between day 10 to 30, whereas control flies live 75 to 90 days [158]. This is mainly due to a dramatically decreased sphingomyelin level at the plasma membrane; as a result, it increases the plasma membrane fluidity and the susceptibility to oxidative stress [158].

#### 4.1.2. Apoptosis

To understand the roles of ceramide in mitochondria-dependent apoptosis, Jain et al. developed a CERT mutant, mitoCERT, that is exclusively localized to mitochondria by replacing the PH domain with outer mitochondrial membrane anchor sequence of A-kinase anchor protein 1 [193]. mitoCERT can deliver ceramide from the ER to mitochondria instead of the *trans*-Golgi [193]. Overexpressing mitoCERT promotes cytochrome c release, activation of caspase-9 and cleavage of PARP, all classic effectors in the intrinsic apoptosis pathway, in HeLa, ovarium carcinoma OVCAR3 and SKOV1, non-small lung carcinoma A549 and colon carcinoma HCT116 cells [193]. This effect is dependent on the ceramide transfer from the ER to mitochondria [193]. The same laboratory further developed a rapamycin-inducible version of mitoCERT, by which they demonstrate that mitochondrial ceramide triggers Bax translocation to mitochondria during apoptosis [194]. These tools well address the pro-apoptotic role of ceramide, but they are not used for interpreting the role of wild-type CERT in apoptosis. Conversely, wild-type CERT is considered anti-apoptotic. Depletion or inhibition of CERT causes accumulation of ceramide in the ER [154,155,160]. Knockdown of CERT predisposes HCT-116 human colon cancer cells, MDA-MB-231 human triple-negative breast cancer cells and A549 human lung carcinoma cells to chemotherapeutic agents-induced ER stress and apoptosis [195]. Meanwhile, CERT ablation sensitizes HER2+ breast cancer cell lines BT474, HCC1954 and SK-BR3 to paclitaxel, doxorubicin and cisplatin-induced cell death [155]. γ-tocotrienol, a member of the vitamin E family, downregulates CERT and induces apoptosis in MIA PaCa-2 pancreatic cancer cells [196]. Gedunin, a limonoid product, inhibits CERT-mediated ceramide extraction from the ER, reduces cellular sphingomyelin level by approximately 50% and exhibits anti-cancer activity in vitro [197,198]. Inhibition of CERT by HPA-12 sensitizes keratinocytes to ultraviolet B-induced apoptosis [199]. In TNFα and anisomycin-induced apoptosis, CERT dissociates from the Golgi due to the Golgi disassembly, followed by cleavage by active caspase-2, 3 and 9 [177]. This also occurs in palmitate-induced apoptosis [156].

#### 4.1.3. Senescence and Mitosis

In murine embryonic cells, CERT deficiency causes accumulation of glucosylceramide instead of ceramide in the ER and mitochondria [161]. This leads to the ER stress, mitochondrial dysfunction, overproduction of reactive oxygen species, increased mitophagy, eventually resulting in G1 phase arrest and premature senescence phenotype in cells [161]. Furthermore, downregulation of CERT is associated with mitochondrial dysfunction and oocyte ageing [200]. Knockdown of CERT by siRNA, inhibition of CERT by HPA-12 or treatment with C6-ceramide induces a mitotic arrest in HCT-116 cells in the presence of paclitaxel [195]. Knockdown of CERT results in mitotic catastrophe in a lysosomal membrane-associated protein 2-dependent manner in paclitaxel-treated HCT-116 cells [155].

#### 4.1.4. Akt Signaling

There are some conflicting reports on the regulation of Akt signaling by CERT. Knockdown of CERT reduces sphingomyelin level by 44% in MCF-7 cells [159]. However, in contrast to the presumably pro-survival role of sphingomyelin, knockdown of CERT and reduction in sphingomyelin improves epidermal growth factor-triggered Akt and ERK activation, which promotes cancer cell migration and focal clustering [159]. In support of this, sphingomyelin synthase 2 deficiency decreases sphingomyelin levels at the plasma membrane, which improves insulin resistance in diet-induced obese mice [201]. Against the above evidence, inhibition of CERT by HPA-12 suppresses Akt translocation and activation at the plasma membrane, sensitizing paclitaxel-induced apoptosis in HCT-116 human colon cancer cells [195]. In addition, downregulation of CERT protein level is associated with insulin resistance in palmitate-treated C2C12 murine skeletal muscle cells and the gastrocnemius muscle isolated from mice on a high-fat diet for 12 weeks [156]. Knockdown of CERT impairs insulin signaling in the presence of palmitate-induced lipotoxicity, whereas overexpression of CERT reduces ceramide level by 4.1-fold and restores insulin sensitivity in palmitate-treated C2C12 myoblasts [156].

#### 4.1.5. Golgi Disassembly and Protein Secretion Sorting

The morphological elongation of the Golgi apparatus is induced at 1 h after treatment with TNFα or anisomycin in HeLa cells, which is followed by structural disruption at 4 h [177]. This change can be abrogated by the pan-caspase inhibitor, indicating it results from apoptosis [177]. The caspase-2, 3 and 9-mediated cleavage of CERT at its D213 inactivates CERT, leading to the Golgi disassembly [177]. In addition, CERT is implicated in the regulation of protein secretion from the Golgi apparatus [164]. Overexpression of CERT elevates protein kinase D activity and augments protein kinase D-dependent protein secretion [164].

#### 4.1.6. Oxidative Stress

Ceramides have long been established to induce reactive oxygen species (ROS) production in mitochondria, leading to oxidative stress [202,203,204,205]. Specifically, C16 ceramide is a major inducer of ROS and oxidative stress, as demonstrated by the differential regulation of ceramide species in CerS2-deleted mouse livers [206]. Sphingomyelin is negatively correlated with membrane fluidity [207]; while increased plasma membrane fluidity is associated with enhanced oxidative stress [208]. Therefore, sphingomyelin may have antioxidant effects, and restricting the conversion of ceramide to sphingomyelin may potentiate oxidative responses. Indeed, knockdown of sphingomyelin synthase 1 sensitizes neuroblastoma Neuro 2A cells to H_2_O_2_-induced oxidative stress and cell death [209], while functionally null nutant of CERT results in oxidative stress in flies [158]. Glutathione peroxidase 8 is a scavenger of ER-derived H_2_O_2_ [210]. Knockdown of this enzyme increases CERT expression, along with a decreased ceramide level and an increased sphingomyelin level in HeLa cells [211]. Meanwhile, the polyunsaturated fatty acid (PUFA) content in phospholipids is decreased, as an adaptive mechanism leading to phospholipid-enriched membranes less prone to oxidative stress [211] and more resistant to free radicals and chemotherapies in cancer cells [212]. This suggests a potential link between CERT and PUFA, which warrants further investigation.

### 4.2. Biological Functions of Long-Form CERT

#### 4.2.1. Neurotoxicity and Neurogenesis

CERT_L_ is highly expressed in neurons of the cerebral cortex, formation of the hippocampus, basal ganglia and the olfactory and nuclei of the thalamus than other regions of the adult rat brain, suggesting it might play a critical role in neuron functions of the central nervous system [213]. CERT_L_ can physically interact with amyloid precursor protein [162]. CERT_L_ inhibits the spontaneous fibrillization of amyloid β in vitro and protects SHSY-5Y human neuroblastoma cells from amyloid β-induced toxicity [162]. CERT_L_ level is reduced in the cortex of mice with Alzheimer’s disease, while ceramide level is elevated in accord [162,214]. Adeno-associated virus-mediated overexpression of CERT_L_ for 12 weeks significantly decreases C16:0 ceramide but increases levels of C16:0, C18:0, C18:1, C20:0, C22:0 and C24:1 sphingomyelin in the cortex, and it ameliorates amyloid β pathology in 5XFAD mice [162]. Administration of mice with CERT inhibitor HPA-12 reverses all of these phenotypes [162]. In addition, CERT can optimize direct membrane trafficking and/or apical membrane signaling in neural stem cells during neurogenesis [215].

#### 4.2.2. Bacterial and Viral Infection

In Chlamydia-infected cells, CERT is recruited to the inclusion membrane, forming the ER-Chlamydial inclusion membrane contact site [216]. This facilitates ceramide transport from the ER to the inclusion membrane [217,218] and the subsequent sphingomyelin synthesis [217,218]. Sphingomyelin is then uptaken by the Chlamydial membrane, which is required for its growth and replication [219]. Knockout of CERT via CRISPR/Cas9 approach decreases Chlamydial infection [220]. In addition, Chlamydia proliferates and forms infective progeny dependent on CERT, but not sphingomyelin synthases [217]. CERT is also implicated in Hepatitis C maturation and secretion [221]. Protein kinase D suppresses Hepatitis C virus secretion by phosphor-inactivating CERT, which can be reversed by overexpression of non-phosphorylatable CERT mutant S132A [221].

#### 4.2.3. Immunological Recognition

Apoptotic cells are cleared by phagocytes, which requires their binding to the complement component C1q [222] via serum amyloid P component [191]. CERT_L_ physically interacts with the serum amyloid P component, and this interaction is inhibited by C1q [191]. Both CERT_L_ and CERT directly bind to C1q, leading to complement activation [192]. The co-localization of CERT_L_ and C1q was also found in apoptotic cells, which might be involved in the recognition of apoptotic cells by C1q [192]. Transgenic overexpression of CERTL in non-lupus-prone mice induces deposition of IgA in the glomerular basement membrane, leading to immune complex-mediated glomerulonephritis [223].

## 5. Role of CERT in Human Cancers

At the cellular level, CERT deficiency is shown to cause the imbalance between ceramide and sphingomyelin, resulting in ER stress, mitochondrial-dependent apoptosis, cell cycle arrest and inactivation of the pro-survival master regulator Akt. These suggest CERT might be a cancer-promoting factor. However, the role of CERT has never been studied in any animal work via genetic manipulation or pharmacological inhibition, and thus the direct evidence is missing. Here we summarize all clinical evidence regarding CERT expression levels in different human cancers (Figure 5), aiming to raise readers’ attention in their future research.

### 5.1. High CERT Expression Associated with Cancer

In support of the hypothesis that CERT promotes cancer, its mRNA level is upregulated by three-fold in ovarian cancer patients after receiving three cycles of neoadjuvant paclitaxel, which is associated with drug resistance [195]. Through an RNA interference functional screening, CERT was identified as one of the 14 genes that confer resistance to paclitaxel in triple-negative breast cancer cells [224]. CERT was also found as a major contributor and an independent predictor of paclitaxel resistance in the MDA cohort and MDA/MAQC II clinical cohorts (GSE16716) of breast cancer [224,225]. High CERT expression is associated with poorer breast cancer outcomes, whereas low CERT expression is correlated to better chemotherapeutic outcomes in a cohort of Leeds Teaching Hospitals, UK [155]. A similar relationship between CERT expression and relapse-free survival was observed through the mining in an online breast cancer database [226]. We searched the information of CERT in Gene Expression Profiling Interactive Analysis 2 (GEPIA 2) via http://gepia2.cancer-pku.cn (accessed on 10 November 2021) [227,228]. As shown in Figure 6, in all 31 types of cancers, the CERT level is significantly higher in pancreatic adenocarcinoma, as compared with the normal tissue.

### 5.2. Low CERT Expression Associated with Cancer

Opposing to the hypothesis that CERT promotes cancer, its mRNA expression is significantly lower in human basal breast cancers compared with non-basal tumors (GSE1561/GSD1329 and GSE2744/GSD2250, *p* < 0.002;) and normal breast tissues (GSE2744/GSD2250, *p* = 0.013) [159,229,230]. CERT protein is also lowly expressed in basal breast cancer cell lines, such as MDA-MB-157, 231, 468 and HS-578T, in comparison with that in luminal breast cancer lines, such as MCF-7, BT474, MDA-MB-453, SK-BR3 and ZR751 [159]. In addition, CERT is lowly expressed in progressing non-muscle invasive bladder cancer as compared with non-progressing tumors obtained from hospitals in Denmark, Sweden, Spain and England [231]. Further, high CERT mRNA expression is correlated with better survival in non-small lung cancer patients in China [232]. In addition, searching in GEPIA 2 (accessed on 10 November 2021), we have found that CERT level is significantly lower in ovarian cancer and urothelial bladder carcinoma, as compared with the normal tissues (Figure 6).

### 5.3. Differential CERT Expression in Different Cancer Contexts

According to the available information, CERT is often found highly expressed in human cancers undergoing chemotherapies [155,195,224], which indicates a role of CERT in drug resistance. CERT mRNA level is significantly increased over 3-fold in ovarian cancer patients after three cycles of neoadjuvant paclitaxel [195]. CERT mRNA expression is also elevated in human colon cancer HCT-116 cells after 8 h treatment with paclitaxel [155]. Mechanistically, a high expression level of CERT reduces ceramide, and thus it alleviates drug-induced ER stress and activates pro-survival PI3K/Akt signaling in ovarian cancer and breast cancer cells [195,224]. The same research group has also identified that deletion of CERT causes an accumulation of ceramide that is required for the initiation of autophagic death in breast cancers treated with chemotherapeutic agents [155]. To better understand the role of CERT in drug resistance, CERT expression should be examined in cancer cohorts with chemotherapies other than paclitaxel, and the comparison should be made between subjects who had successful and failed chemotherapies.

In contrast, low CERT expression is usually reported in cancer cohorts, with no drug treatment information [159,231,232]. In the absence of chemotherapies, knockdown of CERT promotes PI3K/Akt activation in response to epidermal growth factor (EGF), which is associated with sphingomyelin loss at the plasma membrane [159]. In support of this, reducing plasma membrane sphingomyelin improves systemic insulin sensitivity in sphingomyelin synthase 2 knockout mice [201]. Notably, the metabolic benefits are achieved when ceramide levels are increased in peripheral tissues [201]. Opposing to this, CERT improves insulin signaling under lipotoxic stress via decreasing ceramide levels both in vivo and in vitro [156], and ceramide has been well characterized as an inhibitor of Akt signaling [233,234]. Thus, the enhanced EGF-induced Akt signaling upon CERT deficiency [159] and improved insulin sensitivity upon sphingomyelin synthase 2 knockout mice [201] might be attributable to changes in other lipid factors. Indeed, the former study also found a decreased level of cholesterol at the plasma membrane, and the latter one observed a decreased level of circulating FFAs, both relating to Akt signaling.

Based on the current understanding, CERT exerts its functions in cancer primarily through the regulation of ceramide and sphingomyelin. Relative to bioactive ceramide, sphingomyelins mainly serve as building bricks in the cellular membrane and are required for cell growth and proliferation. Sphingomyelin level is difficult to change as drastically as ceramide in cells exposed to insults. This may explain why CERT exhibits a more consistent pro-cancer effect under chemotherapy, as CERT function may be more critical in alleviating the reinforced ceramide accumulation elicited by drug-induced stress. In the absence of chemotherapies, the balance between ceramide and sphingomyelin would be more important to cancer cell biology. However, the cellular functions of sphingomyelin in cancers have not been completely elucidated. Meanwhile, whether the effects of other lipid changes overwhelm the impacts of sphingomyelin and ceramide in specific cancer types remains elusive. Therefore, the differential CERT expression in human cancer cohorts should be further investigated from the following angles, including the stratification of the clinical samples, the CERT studies in primary cancer models, an in-depth understanding of the biological functions of sphingomyelin and a complete overview of lipidome and transcriptome in different cancer contexts.

## 6. Conclusions

Since the identification of the ceramide transport function of CERT by Hanada et al. in 2003, there have been extensive efforts to explore the molecular information, biological functions and human disease relevance of this protein over the past two decades. This is because it directly regulates the balance between sphingomyelin, the most abundant sphingolipid within the cell, and ceramide, the most well-characterized signaling sphingolipid. The structure–function relationship underlying the CERT-mediated ceramide transport from the ER to the *trans*-Golgi has been largely elucidated. This core function of CERT is dependent on its PH domain (Golgi targeting), START domain (ceramide binding) and FFAT motif (ER targeting), and it is tightly regulated by serial phosphorylation in the SRM. The crystal structure of functional domains in CERT has been resolved, which provides precise mechanistic insights. However, the knowledge in the regulation of CERT is currently limited to its post-translational modification. The changes in CERT mRNA levels have been observed in a few human cancers, but how CERT is regulated at the transcriptional level warrants further investigations. At the cellular level, CERT has been implicated in a number of biological functions at a variety of subcellular organelles, including the ER, mitochondria, the nucleus, the Golgi and the plasma membrane. CERT deficiency impairs some essential cellular functions and triggers cell death. Of more importance, the essentiality of CERT is convincingly seen in CERT knockout mice that die at the embryonic stage. Those ceramide-associated cellular impacts of CERT are better understood by virtue of our understanding of ceramide. In contrast, we lack knowledge in the biochemical and biophysical properties of sphingomyelin, which, if improved, can broaden the boundary of the CERT research. Meanwhile, ceramide with different lengths of fatty acyl chains exhibits distinct tissue distribution and cell type-specific biological functions [235,236,237], which might result in differential roles of CERT in different cancer cells. CERT may indirectly regulate other types of lipids, relating to cancer outcomes. An in-depth understanding of this can facilitate us better interpret CERT’s roles in different human cancers. In addition, it is intriguing to know if short-form CERT exerts any biological functions via protein–protein interaction, independent of its lipid transport function. The complete loss of CERT leads to embryonic death, which is presumably the reason for the lack of in vivo studies on the role of CERT in cancer. This overlooked research topic should be investigated using inducible or cell type-specific knockout mice in the future. According to the existing analyses in human cancer samples, a differential expression pattern of CERT is found in different human cancers. Particularly, inconsistent expression levels are seen in breast and ovarian cancers, in which CERT level is downregulated in pooled subjects but upregulated in chemotherapy-resistant patients (Figure 5). Although these are analyzed in different cohorts, they still suggest CERT might play distinct roles in different stages or sub-types of cancers. Therefore, further analyses with human sample stratification and animal research in primary cancer models would contribute to depicting a clearer image of CERT in cancer. Lastly, as nearly all cancer-related CERT functions derive from its regulation of ceramide and sphingomyelin. With a new insight provided by the CERT research, we propose if the ratio between these two lipids is a determinant of cancer development and a predictor of cancer prognosis. In support of this, sphingolipid enzymes, such as sphingomyelin synthases, sphingomyelinases and ceramidases, that can regulate this ratio are all implicated in cancer. With the exponentially increasing applications of lipidomics, this could be answered in the near future. In addition, we should continue to interrogate if CERT expression is binarily correlated to cancer incidence. Following the notion that the ratio of β-amyloid 40 and 42 more accurately diagnoses Alzheimer’s disease, the ratio of CERT to other paired factors may better predict cancer development and prognosis.

In summary, CERT might be a critical factor in cancers. Further investigations into CERT and its related ceramide and sphingomyelin will help to understand the fundamental aspects of cancer biology and identify novel druggable targets for the prevention and treatment of cancer.

## Figures and Tables

**Figure 1 ijms-22-13184-f001:**
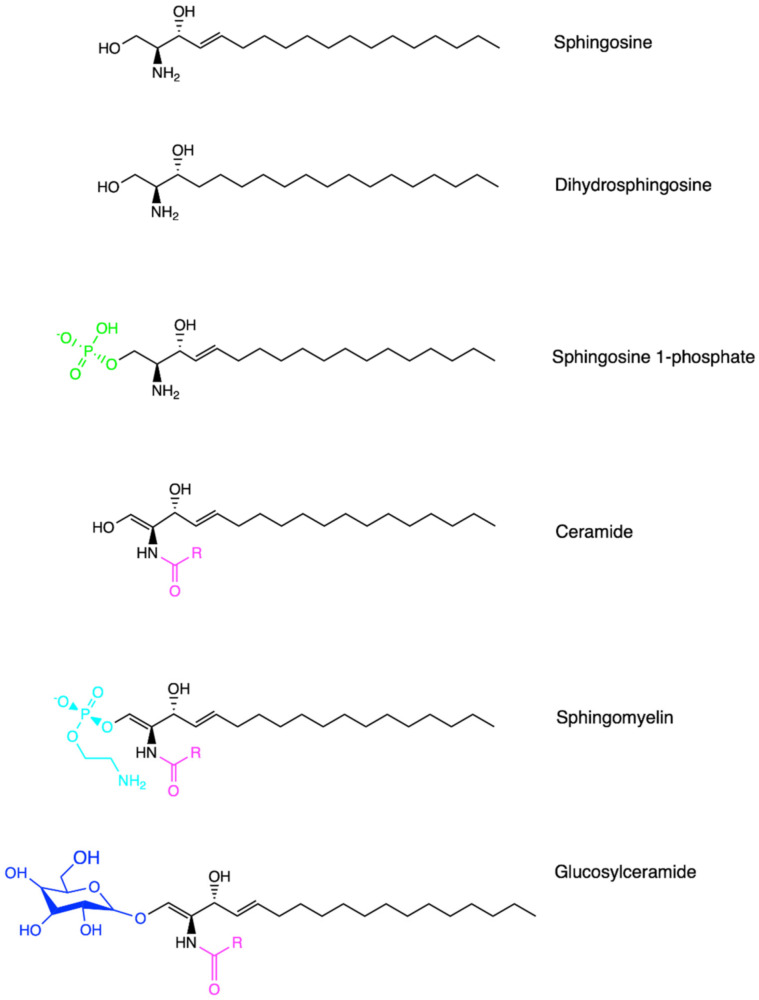
Chemical structures of sphingolipids. Sphingosine represents the common sphingolipid backbone as the sphingoid base. Dihydrosphingosine, also known as sphinganine, possesses a saturated sphingoid backbone. Ceramide contains an additional fatty acyl chain via the N-link to the sphingoid base. Sphingosine 1-phosphate, sphingomyelin and glucosylceramide are generated via O-linked modification from sphingosine or ceramide, respectively. Created in ChemDraw^®^.

**Figure 2 ijms-22-13184-f002:**
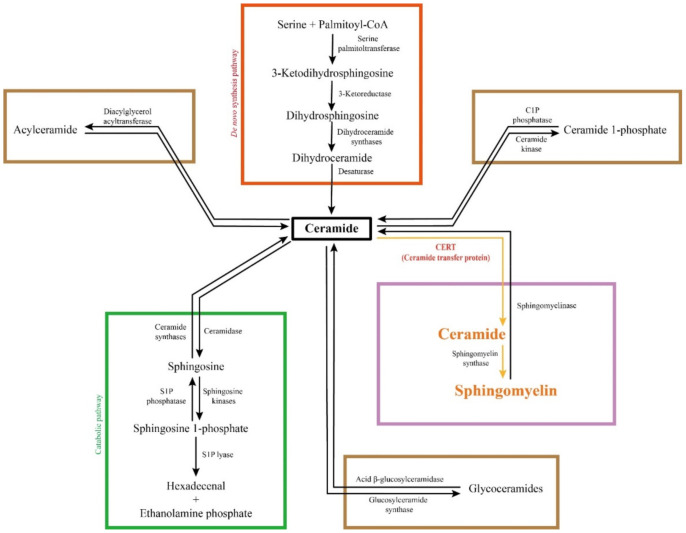
Sphingolipid metabolic network. Ceramide is biosynthesized from amino acid and fatty acid via four steps. It can be degraded in the catabolic pathway into non-lipid products. Meanwhile, ceramide can be converted into complex sphingolipids, such as sphingomyelin, glycoceramides, acylceramide and ceramide 1-phosphate, in a reversible fashion. The majority of ceramide is converted into sphingomyelin, the most abundant sphingolipid species, which is replied on CERT-mediated endoplasmic reticulum to Golgi ceramide transport.

**Figure 3 ijms-22-13184-f003:**
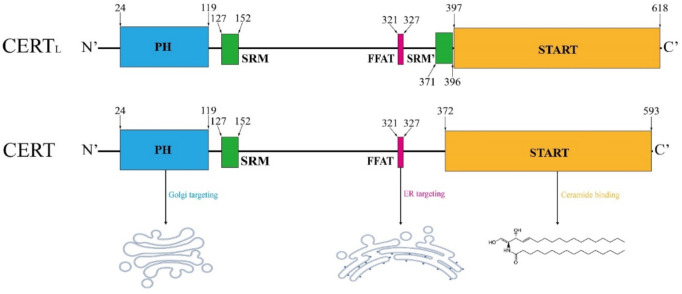
Schematic structure of CERT. The structural elements are colored as follows: Pleckstrin homology (PH) domain (blue), serine-rich motif (SRM, green), two phenylalanines in an acidic tract (FFAT) motif (pink), StAR-related lipid-transfer (START) domain (orange), major phosphorylation sites (red). The short-form CERT lacks a SRM in the proximity of START domain, as compared with the long-form variant. Amino acid sequence is adapted from [134].

**Figure 4 ijms-22-13184-f004:**
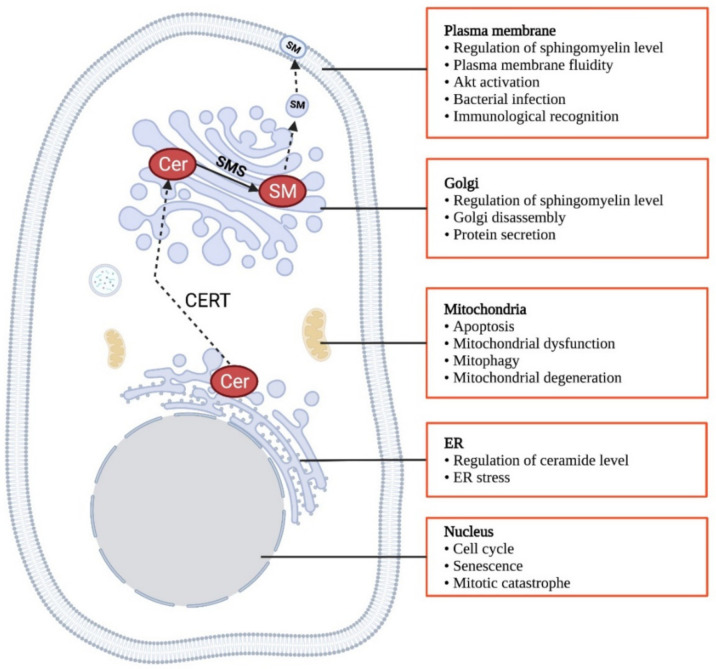
Biological functions of CERT. Created in Biorender.com (accessed on 11 November 2021).

**Figure 5 ijms-22-13184-f005:**
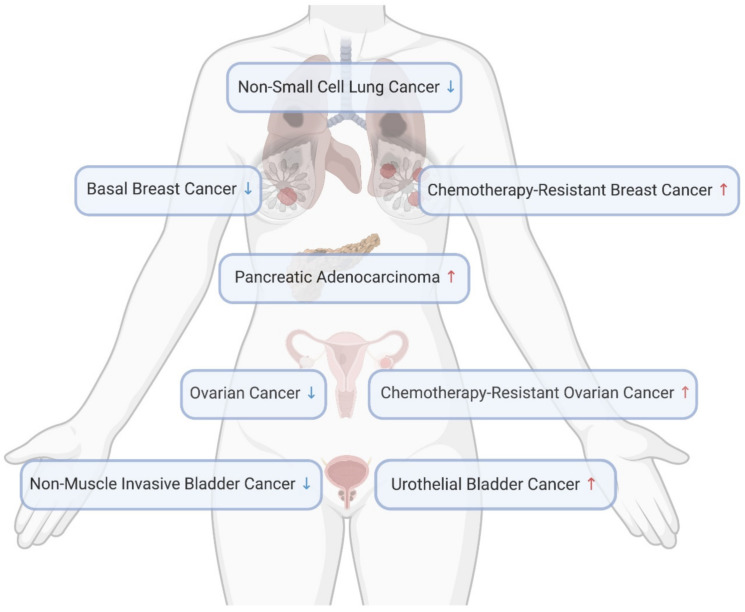
CERT level changes in cancers. Created in Biorender.com (accessed on 11 November 2021).

**Figure 6 ijms-22-13184-f006:**
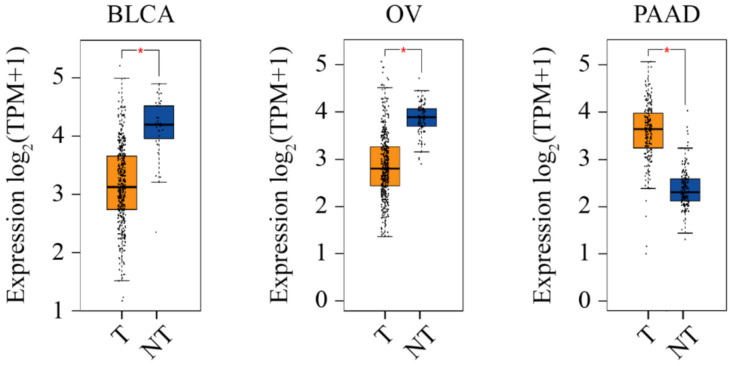
CERT expression level in three cancers. On the GEPIA 2 platform, the CERT expression level is analyzed using expression DIY with ANOVA (Log2FC = 1, q-value cutoff = 0.01) and the matching of TCGA and GTEx data. The * symbol indicates the statistical significance. BLCA, urothelial bladder carcinoma; OV, ovarian cancer; PAAD, pancreatic adenocarcinoma; TPM, transcripts per million. The graph is adapted from GEPIA 2 and edited in Adobe Illustrator.

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
