# Peer review of "Ceramide Transfer Protein (CERT): An Overlooked Molecular Player in Cancer"

_ijms, 2021, doi:10.3390/ijms222413184_

Round 1

Reviewer 1 Report

This review paper summarizes in detail that CERT is closely related to cancer development. I think a major revision is needed prior to publication. It can be seen that overexpression or underexpression of CERT is closely related to the occurrence of various cancers. However, there is a lack of explanation on how the mechanism of cancer caused by CERT underexpression is different from that of cancer caused by CERT overexpression. The authors' sharp insight into why CERT mRNA overexpression/underexpression in specific tissues is associated with cancer development should be provided. Furthermore, data or insight about the change in CERT mRNA level according to chemotherapy is needed. If the chemotherapy is successful, will the CERT mRNA level gradually return to the normal cell level? Is it really the best approach to binaryly correlate CERT expression level with cancer incidence?The ratio of CERT expression level and other molecular level may be more accurate in predicting cancer incidence. (For example, in the case of diagnosing Alzheimer's dementia, only the beta-amyloid expression level is inaccurate, but using the ratio of beta-amyloid(40) to beta-amyloid(42) makes the diagnosis more accurate, and the ratio with the tau expression level is even more accurate.)

Reviewer 2 Report

The manuscript is very interesting. The authors propose the role of Ceramide transfer protein in cancer. The structure of the manuscript is adequate and consistent with the purpose of the review. Each section of the review is very well supported. The bibliography used is adequate and up-to-date. The figures are good and relevant to the purpose of the review. However, I have the following comments.

I. Minor comments:
1. Improve the wording of the objective of the manuscript.
2. Ceramide transfer protein (CERT) may be related to the regulation of transcription factors PPARs. PPARs are regulated by polyunsaturated fatty acids (PUFAs). If so, it would be interesting to briefly discuss this point.
3. I suggest to briefly discuss the role of ceremides in oxidative stress and cell damage.

Round 2

Reviewer 1 Report

The manuscript is well revised far enough for publication at this journal.